# Highly stable QLEDs with improved hole injection via quantum dot structure tailoring

Weiran Cao[1], Chaoyu Xiang[1], Yixing Yang[1], Qi Chen[2], Liwei Chen ⓘ [2], Xiaolin Yan[1] & Lei Qian[1]

For the state-of-the-art quantum dot light-emitting diodes, while the ZnO nanoparticle layers can provide effective electron injections into quantum dots layers, the hole transporting materials usually cannot guarantee sufficient hole injection owing to the deep valence band of quantum dots. Developing proper hole transporting materials to match energy levels with quantum dots remains a great challenge to further improve the device efficiency and operation lifetime. Here we demonstrate high-performance quantum dot light-emitting diodes with much extended operation lifetime using quantum dots with tailored energy band structures that are favorable for hole injections. These devices show a $T_{95}$ operation lifetime of more than 2300 h with an initial brightness of 1000 cd m$^{-2}$, and an equivalent $T_{50}$ lifetime at 100 cd m$^{-2}$ of more than 2,200,000 h, which meets the industrial requirement for display applications.

---

[1] TCL Corporate Research, Nanshan District, No. 1001 Zhongshan Park Road, Shenzhen 518067, China. [2] i-Lab, CAS Center for Excellence in Nanoscience, Suzhou Institute of Nano-Tech and Nano-Bionics, Chinese Academy of Sciences, 398 Ruoshui Road, Suzhou Industrial Park, Suzhou 215123, China. These authors contributed equally: Weiran Cao, Chaoyu Xiang. Correspondence and requests for materials should be addressed to X.Y. (email: yan-xl@tcl.com) or to L.Q. (email: qianlei@tcl.com)

Quantum dot light-emitting diodes (QLEDs) that electrically excite quantum dots (QDs) can fully exploit their unique properties, such as narrow emission spectra, size-controlled emission wavelength, high quantum yield and inherent stability[1–7]. Ever since the first demonstration in 1994, performance of QLEDs, especially the electroluminescence efficiency of device, have been dramatically boosted from less than 0.1% to more than 20%, owning to well-controlled quantum dot materials, suitable device architectures and optimized fabrication processes. However, the half lifetime ($T_{50}$) of QLEDs was only in the range of 100,000 h with an initial brightness of 100 cd m$^{-2}$, which is far below the requirement of display applications.[1, 8, 9] In the state-of-art organic/inorganic hybrid QLEDs, while commonly used ZnO can provide effective electron injection, the relatively large energy level offset between QDs and hole transport layer usually impedes hole injection, leading to unbalanced electrons and holes and hence low device efficiency.[8–10] Moreover, exceeding carriers accumulate at the barrier interface during the device operation, not only acting as the non-radiative centers but also rising the driving voltage and limiting device lifetime.[11–13] Though finding hole transport materials (HTMs) to match the energy levels of QDs is likely to improve hole injections, it is a great challenge to design organic HTMs with deep enough energy levels to match the deep valance band of QDs.[14–16] Other efforts from our previous and other studies have been made on electron transport properties to deal with this problem by either tuning the QD conduction band or impeding electron injection. Even though all of them greatly improved the QLED performance, the operation lifetime of these devices was still not good enough for display applications.[8, 9] To solve the lifetime issue, we tackle the hole barrier issue directly by tailoring the band energy levels of QDs to reduce the injection barriers and improve device performance. A high-quality colloidal QD usually comprises of an inorganic semiconductor core and a semiconductor shell with wider energy bandgap to passivate dangling bonds of core surface and confines electron and hole wavefunctions for good luminescent properties and reliability[17–19]. However, a larger energy bandgap shell limits the injection of carriers in electrical excited devices, due to the relatively shallower highest occupied molecular orbitals (HOMOs) of the most widely used organic HTMs in QLEDs[10, 14–16]. Instead of searching for proper HTMs, it is favorable to design energy band structure of quantum dots to make better energy level alignment with HTM in QLEDs, since the energy level tunability is one of the promising advantages of QDs.

Here we demonstrate a favorable energy level alignment in QLEDs by utilizing core/shell quantum dots with low bandgap shell. With the reduction in injection barrier from the HTM to quantum dot layer in QLEDs, less charges accumulate at the HTM/QDs interface, leading to much extended operation lifetime. A $T_{95}$ operation lifetime of more than 2300 h with an initial brightness of 1000 cd m$^{-2}$ is successfully achieved, which meets the industrial requirements of display and makes QLED technology truly towards the commercialization.

## Results

**Characterization of quantum dots.** Following the energy design strategy, we synthesized CdSe/Cd$_{1-x}$Zn$_x$Se/ZnSe QDs (ZnSe-QDs) with low bandgap ZnSe shell which allows efficient hole injection. The high photoluminescence efficiency of them is preserved by optimizing the composition gradient and thickness of the shell. Conventional CdSe/Cd$_{1-x}$Zn$_x$Se$_{1-y}$S$_y$/ZnS QDs (ZnS-QDs) that have the same core as ZnSe-QDs but with a wider bandgap ZnS outer shell were also prepared for comparison. Inserted in Fig. 1a are transmission electron microscopy (TEM) images of ZnSe-QDs and ZnS-QDs, respectively. While the average size of ZnS-QDs are measured to be about 8.4 nm, the optimized ZnSe-QDs show a larger size of about 9.8 nm. Both ZnSe-QDs and ZnS-QDs have gradual changes in composition from cores to shells (Supplementary Fig. 1), and the size and composition of both QDs were optimized for achieving the best device performance. Due to the reduced synthesis complexity with only three elements, the ZnSe-QDs have a better size and composition distribution (Supplementary Fig. 2), leading to a much narrower full-width at half-maximum (FWHM) of 21 nm, compared to that of 28 nm for the ZnS-QDs with the four element reaction system, as shown in Fig. 1a. It is also found that, although ZnSe has narrower energy bandgap, assumptively less efficient confinement than ZnS, the ZnSe-QDs can still achieve similarly high photoluminescence quantum yield (PLQY) of 84%, compared to that of ZnS-QDs (PLQY = 87%), by precisely optimizing the gradient of the intermediate shell (Cd$_{1-x}$Zn$_x$Se) and the thickness of the outer ZnSe layer for efficient wavefunction confinement[18, 20] (Supplementary Fig. 3).

We used high hole mobility poly(9,9-dioctylfluorene-co-N-(4-(3-methylpropyl))diphenylamine) (TFB) as HTM in our devices. The injection barriers from the TFB to QD layers for the ZnS-QD and ZnSe-QD devices are derived from ultraviolet photoelectron spectroscopy (UPS) and scanning Kelvin probe microscopy (SKPM) measurements using reported methods[21, 22]. The UPS reveals the energy level depth of individual layers (Supplementary Fig. 4) and the SKPM provides the interfacial dipoles that correspond to the vacuum level shifts (Supplementary Fig. 5)[23–27]. From them the energy level alignments between layers were obtained. As shown in Fig. 1b, the valence band offset at the TFB/ZnS-QD interface is calculated to be about

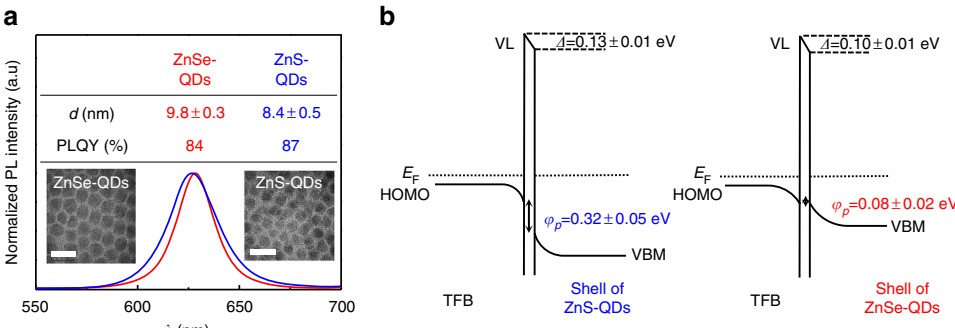

**Fig. 1** Colloidal quantum dots. **a** Photoluminescence spectra of the ZnS-QDs and ZnSe-QDs. Inset: TEM images and a summary of diameter (*d*), and photoluminescence quantum yield (PLQY) of ZnS-QDs and ZnSe-QDs. **b** Energy band diagrams with Fermi-level alignment at TFB/ZnS-QD and TFB/ZnSe-QD interfaces. Here, only the energy levels of the shells of QDs are presented

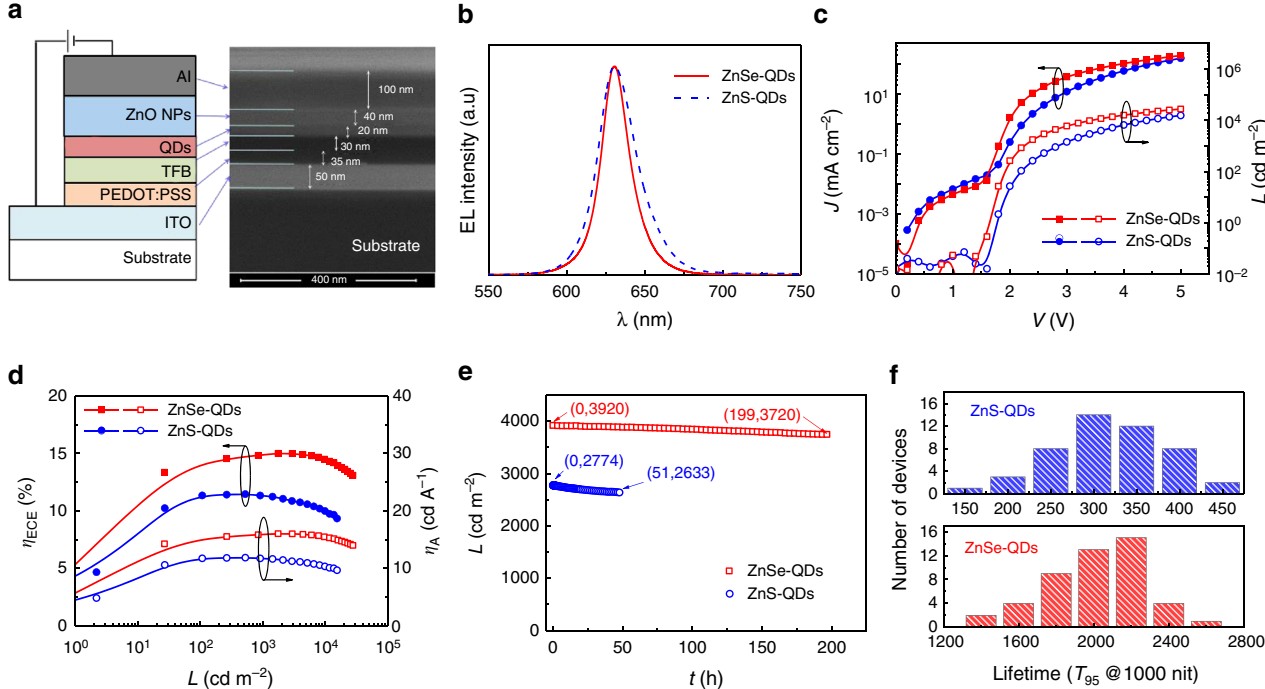

**Fig. 2** Device performance. **a** Device structure and cross-section scanning electron microscopy (SEM) image with layer thicknesses labeled. **b** Electroluminescence spectra of the ZnSe-QD and ZnS-QD devices. **c** Current density-luminance-voltage (J-L-V) characteristics for the two devices. **d** Current efficiency ($\eta_A$) and external quantum efficiency ($\eta_{EQE}$) as a function of luminance for the two devices. **e** Stability data (luminance vs. time) for the two devices. **f** Histograms of $T_{95}$ lifetime with initial luminance of 1000 cd m$^{-2}$ measured from 48 ZnSe-QD devices and 48 ZnS-QD devices

$0.32 \pm 0.05$ eV, whereas that at the TFB/ZnSe-QD interface is negligible, which proves that this excellent alignment of the valance band and HOMO between ZnSe-QDs and TFB interface promises with a significantly reduced hole injection barrier in the corresponding QLEDs.

**Device performance of QLEDs based on ZnSe-QDs and ZnS-QDs.** The layered QLED devices (Fig. 2a) were all spin coated except the two electrodes (see Methods section for detailed device fabrication procedures). The normalized electroluminescence (EL) spectra of the devices based on ZnSe-QDs and ZnS-QDs are shown in Fig. 2b. Both EL spectra have a red-shift emission peak at 631 nm, compared to the photoluminescence spectra of QDs in solution. The ZnSe-QD-based device maintains the narrow emission of ZnSe-QDs, with FWHM of 21 nm, while the FWHM of ZnS-QD device is slightly broadened to 30 nm. Figure 2c compares the current density-luminance-voltage (J-L-V) characteristics of QLEDs based on ZnSe-QDs and ZnS-QDs. The ZnSe-QD device shows a turn-on voltage ($V_T$, defined as the required operation voltage for achieving the luminance of 1 cd m$^{-2}$) of 1.7 V and a required voltage of $V = 2.3$ V for achieving 1000 cd m$^{-2}$ luminance (red squares in Fig. 2c), whereas the ZnS-QD device has a slightly higher $V_T = 1.8$ V and a much higher $V = 2.9$ V at the luminance of $L = 1000$ cd m$^{-2}$ (blue circles in Fig. 2c), suggesting the reduction of injection barrier by the ZnSe shells in ZnSe-QDs. The external quantum efficiency ($\eta_{EQE}$) and current efficiency ($\eta_A$) as a function of the luminance ($L$) of the ZnSe-QD and ZnS-QD devices are exhibited in Fig. 2d and the efficiency statistics of these devices are shown in Supplementary Fig. 6. The ZnSe-QD device shows the maximum $\eta_{EQE}$ and $\eta_A$ value of 15.1% and 15.9 cd A$^{-1}$ (red squares in Fig. 2d), respectively, which is about 50% higher than that of the ZnS-QD device ($\eta_{EQE} = 11.4\%$ and $\eta_A = 11.8$ cd A$^{-1}$) (blue circles in Fig. 2d). The optimized device performances of different ZnSe-QDs are presented in Supplementary Fig. 3.

Benefiting from the smaller injection barrier, the ZnSe-QD-based devices show remarkably high operation lifetime. The comparison of luminance vs. time for a ZnSe-QD device and ZnS-QD device is presented in Fig. 2e. Both devices were driven at a constant current density of 25 mA cm$^{-2}$ under ambient conditions with the humidity of 30–40% and temperature of 21–23 °C. Initially, the luminance of the ZnSe-QD device and ZnS-QD device was $L = 3920$ cd m$^{-2}$ and $L = 2774$ cd m$^{-2}$, respectively, slowly decreasing over operation time. While the luminance of the ZnS-QD device drops to 95% of its initial value ($L_{95\%} = 2633$ cd m$^{-2}$) after about 50 h (blue circles in Fig. 2e), it takes more than 200 h for the ZnSe-QD device reaching its 95% luminance value ($L_{95\%} = 3720$ cd m$^{-2}$) (red squares in Fig. 2e). The $T_{95}$ lifetime at an initial brightness of 1000 cd m$^{-2}$ ($T_{95}$ @ 1000 cd m$^{-2}$) can be extrapolated to 2320 h and 320 h for the ZnSe-QD device and ZnS-QD device, respectively, with acceleration factor of $n = 1.80$ by fitting $T_{95}$ values at various luminance values (Supplementary Fig. 7). As shown in the histograms of the lifetime of 48 ZnSe-QD devices and ZnS-QD devices, respectively, in Fig. 2f, these devices show great reproducibility and the ZnSe-QD-based devices exhibit an average $T_{95}$ lifetime at 1000 cd m$^{-2}$ of over 2000 h. For a direct comparison with previously reported values of QLED lifetime, a half ($T_{50}$) lifetime with initial luminance of 100 cd m$^{-2}$ can be estimated to be over 2,260,000 h (Supplementary Fig. 7), which is about one order of magnitude higher than previous reported values[8, 9]. The operation lifetime we reported here meets the requirement of display application, which is another milestone on the way of commercialization of QLED technology.

**Mechanism study for the long operation lifetime.** We conducted J-L-V sweeps during accelerated lifetime tests to study the degradation mechanism of devices. Both the ZnSe-QD and ZnS-QD devices were operated at a constant current density of 50 mA cm$^{-2}$, and the normalized luminance decay vs. time are shown in

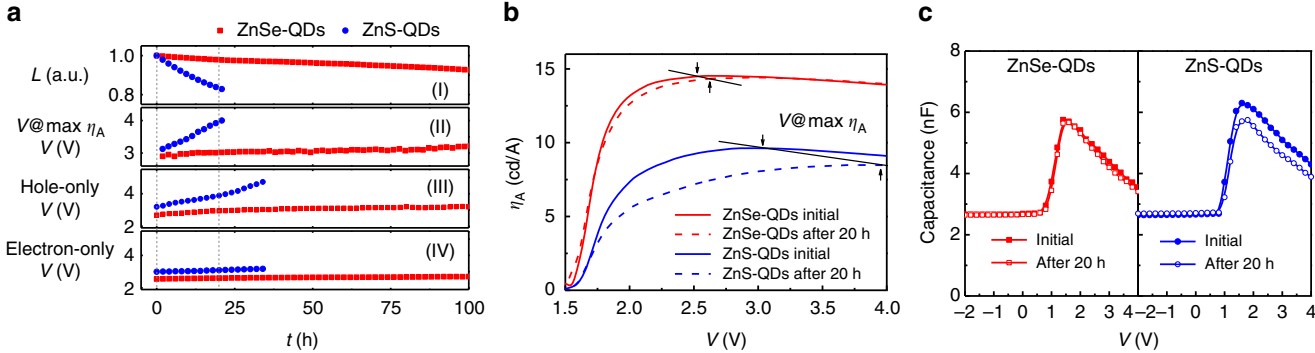

**Fig. 3** Degradation mechanism. **a**, Normalized luminance ($L$) and operation voltage at maximum $\eta_A$ (V@ max $\eta_A$) vs. time for the ZnSe-QD and ZnS-QD devices operated at a constant current, and operation voltage vs. time of hole-only and electron-only ZnSe-QDs and ZnS-QDs based devices operated at a constant current. **b** Current efficiency ($\eta_A$) as a function of voltage for the ZnSe-QD and ZnS-QD devices measured before the lifetime test and after 20 h continuous operation. **c** Capacitance vs. voltage for the two devices measured before the lifetime test and after 20 h continuous operation

Fig. 3a. Figure 3b compares the device efficiency as a function of driving voltage for these ZnSe-QD and ZnS-QD devices before the lifetime tests and after 20 h of continuous operation, respectively. For both devices, the changes in turn-on voltage are negligible, whereas the required voltages for these devices reaching maximum efficiency (noted as $V$@max $\eta_A$), which are considered as the optimum charge balance points[11], are far different. As plotted in Fig. 3a, while the voltage at maximum efficiency quickly increases from 3.0 V to over 4.0 V in 20 h for the ZnS-QD-based device (blue circles in Fig. 3a), the optimum charge balance point for the ZnSe-QD device only slightly shifts about 0.3 V to higher voltage in 100 h (red squares in Fig. 3a), suggesting that the ZnSe-QD device has more balanced electron and hole currents over time (also see single-carrier $J$-$V$ characterization in Supplementary Fig. 8). The QD films based on ZnS-QDs and ZnSe-QDs exhibit similar surface roughness according to the atomic force microscopy (AFM) images (Supplementary Fig. 9), and therefore the influence of film qualities on such voltage changes can be eliminated. It is believed that charges accumulate at the interfaces owning to the injection barriers, leading to the decrease in effective charge transport and hence the rise of operation voltage[11, 28]. As compared in the capacitance-voltage plots (Fig. 3c), the peak capacitance dropped about 1 nF for the ZnS-QD device after 20 h of operation, whereas there was no significant difference observed for the ZnSe-QD device, suggesting that more traps formed during the operation and affected the charge transport due to the relatively larger barrier of ZnS shell in the ZnS-QD device[12, 29]. To distinguish whether the hole or electron barrier dominates the degradation, voltage changes in single-carrier devices under continuous operation (Fig. 3a) are measured, where the single-carrier devices only allow the transport of one kind of carriers (electrons or holes) during operation[6, 30]. Only slight voltage growth was observed during the operation for both the ZnSe-QD and ZnS-QD electron-only devices, suggesting favorable and steady electron transport with time in both devices. However, the situation for the hole-only devices is far different. While the ZnSe-QD-based device showed almost unchanged voltage with time (red squares in Fig. 3a), a quick operation voltage rise for the ZnS-QD-based hole-only device was observed (blue circles Fig. 3a). The magnitude of voltage rise with time in the hole-only devices is comparable to the optimum charge balance point changes in the dual-carrier devices, indicating that the hole behaviors are the major factor affecting the operation lifetime. In ZnSe-QDs, the shallower valance band of ZnSe allows more efficient hole injection from the hole transport

layer, compared to that of the ZnS shells of ZnS-QDs, leading to less charge accumulation at the interface and thereby less trap formation over time in the device.

## Discussion

The present work demonstrates highly stable QLEDs whose operation lifetime meets the requirement of display industry. This is achieved through solving the long-standing hole injection problem by tuning the energy level of QDs to match the adjacent layers in QLED, instead of searching for transport materials with proper energy levels. By tailoring energy band structure through gradient composition and shell thickness, high photo-luminescence quantum yield QDs with ZnSe outer shell were successfully synthesized, which exhibits reduced carrier injection barriers, especially for that of the holes. QLEDs based on the ZnSe-QDs show a $T_{50}$ lifetime of over 2 million hours at 100 cd m$^{-2}$, which is about one order of magnitude higher than previous reports (Supplementary Fig. 10). We believe that such concepts to improve the operation lifetime along with energy level design strategies of QLEDs will open minds for other QD-based optoelectronic devices, which would lead to the realization for full-color QLED displays in the future.

## Methods

**Material synthesis.** The quantum dots used in this work were prepared according to the methods previously reported in the related literatures with appropriate modifications[9, 31, 32]. For a typical synthesis of CdSe/Cd$_{1-x}$Zn$_x$Se$_{1-y}$S$_y$/ZnS QDs (ZnS-QDs), 0.4 mmol of CdO, 6 mmol of zinc acetate, and 7 ml of oleic acid were placed in a 50 ml flask and heated to 170 °C in flowing high-purity argon for 30 min. Then, 15 ml of 1-octadecene was added to the flask and the temperature was elevated to 300 °C. A stock solution containing 0.9 mmol of selenium (Se) dissolved in 2 ml of trioctylphosphine (TOP) was quickly injected into the flask. The reaction temperature was kept for 10 min. Then, 0.1 mmol Se and 0.3 mmol sulfur (S) dissolved in 1 ml TOP were injected into the flask at the elevated temperature of 300 °C and reacted for 10 min. Finally, the reaction mixture was cooled to room temperature. The resulting QDs were precipitated with acetone for several times and finally dispersed in octane. For the synthesis of CdSe/Cd$_{1-x}$Zn$_x$Se/ZnSe QDs (ZnSe-QDs), similar procedure is performed, except for the absence of S precursor addition and necessary modification of shell growth. The shell thickness of ZnSe was controlled by the amount of Se in the second injection.

ZnO nanoparticles were synthesized by a solution-precipitation process reported in the literature[5, 33]. For a typical synthesis, a solution of zinc acetate in dimethyl sulfoxide (0.5 M) and 30 ml of a solution of tetramethylammonium hydroxide in ethanol (0.55 M) were mixed and stirred for 1 h in ambient conditions, then washed and dispersed in ethanol for device fabrication.

**Device fabrication.** QLED devices were fabricated by consecutively spin coating each layer on glass substrates pre-coated with an indium-tin oxide anode (ITO, sheet resistance is about 50 Ω □$^{-1}$). Prior to use, all substrates were cleaned by ultrasonic treatment in tergitol/deionized (DI)-water solution, DI-water, and

isopropanol, respectively, followed by treating under ultraviolet (UV)–ozone for 15 min. A 35 nm thick PEDOT:PSS (Baytron P VP AI 4083) layer was first spin coated onto the cleaned substrates at 5000 r.p.m. for 60 s and annealed at 150 °C in air to remove residual water. A 30 nm thick TFB (American Dye Source) layer was then deposited onto the PEDOT:PSS layer by spin coating from its chlorobenzene solution (8 mg ml$^{-1}$, 3000 r.p.m.) in a N$_2$ glove-box, followed by a thermal anneal at 150 °C for 30 min. After that, a QD layer and a ZnO nanoparticle layer were sequentially spin coated from the octane and ethanol solution in the glove-box, respectively, and annealed in the same environment at 120 °C to remove the residual solvents. Thicknesses of the QD layer and ZnO layer were varied by changing the solution concentrations and spin speeds. The optimal thicknesses for QD layers and ZnO layers were 20 nm (spin coated from solutions with a concentration of 15 mg ml$^{-1}$ at 2000 r.p.m. for 60 s) and 40 nm (spin coated from solutions with a concentration of 30 mg ml$^{-1}$ at 3000 r.p.m. for 60 s), respectively. After the deposition of the solution-processed layers, all samples were transferred to a vacuum deposition chamber with chamber pressure less than 10$^{-6}$ torr ($P < 10^{-6}$ torr) for Al cathode (100 nm thick) deposition, followed by the final encapsulation with a UV-curable epoxy and cover glasses in the N$_2$ glove-box. All the devices had the emitting area of 4 mm$^2$ that was defined by the overlapping of ITO and Al electrodes.

**Characterizations**. Photoluminescence spectra of QDs were obtained in hexane solutions with an Edinburgh FS5 steady-state fluorescence spectrometer by excitation at the absorption maxima. The absolute quantum yield of QD solutions was measured using the built-in integrating sphere of Edinburgh spectrometer. TEM images were taken using a JEOL-2100F TEM (Shanghai Your Equipment Technology Co. Ltd) with 200 keV electron beam energy and X-ray diffraction patterns were taken using a Buker D8 Advance Diffractometer (Shanghai Your Equipment Technology Co. Ltd). Electroluminescence spectra were obtained using an Ocean Optics USB 2000+ spectrometer with the devices driven at a constant current with a Keithley 2400 source meter. The J-L-V characteristics of the devices were taken under ambient conditions with a Keithley 2400 source meter measuring the sweeping voltages and currents and a Keithley 6485 Picoammeter together with a calibrated silicon detector (Edmund) measuring light intensities. Luminance was calibrated using a photometer (Spectra Scan PR655) with the assumption of Lambertian emission pattern of all devices[34]. The lifetime test were conducted under ambient conditions using a commercialized lifetime test system (Guangzhou New Vision Opto-electronic Technology Co. Ltd.). The UPS was collected on ULVAC-PHI 5000 Versaprobe II (ULVAC-PHI Inc.). Samples were handled with care and the air exposure time before being transferred to vacuum chamber was less than 10 s. The SKPM was carried out on a Park XE-120 AFM (Park Systems Corp., Suwon, Korea) placed in a N$_2$-filled glove-box. Pt-coated conducting tips (OMCL-AC240TM, Olympus Corp., Tokyo, Japan) with a resonance frequency of ~70 kHz and a spring constant of ~2 N m$^{-1}$ were used for a two-pass scan amplitude modulation SKPM to measure the surface potential. During the first pass, standard AC mode imaging (typical tip oscillation amplitude 20 nm) was performed to acquire the topography and phase signals of the sample; in the second pass, the tip was lifted up by a certain height (typically 10 nm) and scanned on the basis of the topography line obtained from the first pass. An AC voltage (1 V in amplitude at the tip resonant frequency) was applied between the cantilever and sample, and the DC voltage required at the tip that nullifies the tip-sample interaction is collected as the surface potential signal.

**Data availability**. The data that support the findings of this study are available from the corresponding author upon reasonable request.

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

## Acknowledgements

This work is financially supported by Ministry of Science and Technology of the People's Republic of China (2016YFB0401604, 2016YFA0200703), Guangdong Provincial Department of Science and Technology (2015B090913001), Shenzhen Innovation of Science and Technology Committee (JSGG20160818152648289), the CAS Research

Equipment Development Program (YZ201654), and the National Natural Science Foundation of China (Grant Nos. 2162304 and 11504408). The authors acknowledge Shenzhen China Star Optoelectronics Technology Co., Ltd for supporting high-quality ITO substrates. The authors also gratefully acknowledge H. Qin, C. Yang, and J. Qiu for assistance with quantum dot synthesis, W. Wang, Z. Liang, L. Li, Z. Xin, P. Zhu, D. Zhang, and M. Xie for assistance with device fabrication, and J. Liu, L. Li, T. Zhang, and C. Wang for assistance with data collection.

## Author contributions

W.C., C.X. and Y.Y. supervised the material synthesis, device fabrication, and data collection for the paper. L.Q., W.C., C.X. and Y.Y. discussed results, designed tests, and postulated mechanisms to explain the performance of QLED. Q.C. and L.C. conducted the UPS and SKPM measurements and analyzed the results. W.C. wrote the first draft of the manuscript and C.X., Y.Y., Q.C., L.C. and L.Q. provided suggestions and revisions. L. Q. and X.Y. supervised the work and specified the directions of the QLED research in TCL.

## Additional information

**Competing interests:** The authors declare no competing interests.

