## [Peer Review File · Nature Communications]

Reviewers' comments:

Reviewer #1 (Remarks to the Author):

In this work, Cao et al. presents a new strategy for improving hole injection by synthesizing CdSe/Cd_{1-x}Zn_xSe/ZnSe QDs (ZnSe-QDs) with low bandgap ZnSe shell instead of conventional shell of wider bandgap ZnS (ZnS-QDs). They demonstrated that the energy barrier between organic hole-transport layer (HTL) of TFB and ZnSe-QDs is lower compared to the case of the ZnS-QDs, resulting in improved hole injection in quantum dot light emitting diodes (QLEDs). As a result of improved hole injection, the QLEDs with ZnSe-QDs exhibited a maximum EQE > ~ 15% and a T95 operation lifetime > 2,300 hours at an initial brightness of 1,000 cd/m². This is a remarkable achievement toward developing stable QLEDs for practical displays. Therefore, I recommend its publication after addressing the following issues.

- (1) Although the authors report record device lifetime, there is not a deep investigation on physical processes that can lead to this improvement and the degradation mechanisms for QLEDs. The authors claim that the hole injection and transport is the major factor affecting the operation lifetime for QLEDs mainly based on faster voltage rise for the ZnS-QDs based hole-only device compared to the ZnSe-QDs based hole-only device (Fig. 3 a). Even though the HTL/QD interfacial properties play a role for device degradation, there are other possible important factors, for example, unbalanced charge injection can increase non-radiative Auger-type recombination, or ZnS-QDs may have more surface defects compared to ZnSe-QDs, etc. Moreover, the QLEDs and hole-only device have different interface at the cathode. Thus, the authors are recommended to elaborate more how the ZnSe-QD-based device shows such a large enhancement in the operation lifetime.
- (2) According to the energy band diagrams with Fermi-level alignment at TFB/ZnS-QDs and TFB/ZnSe-QDs interfaces in Fig. 1, the CBM of the ZnSe-QDs appears to be higher than that of the ZnS-QDs. Moreover, the shell thickness of ZnSe-QDs is larger because the core size of two QDs is the same and the diameter of the ZnSe-QDs is larger ($d \sim 9.8$ nm compared to 8.4 nm for ZnS-QDs). Thus, it is expected that less current flows in the electron-only device based on ZnSe-QDs. However, the Extended Data Figure 8 shows higher current flow for the electron-only device with ZnSe-QDs. Thus, morphology of QDs layer may be different for each case. The presence of pinholes or voids in such a thin (~ 20 nm) QDs film may affect the current density for electron-only as well as hole-only devices. Thus, the authors should address the thin film morphology difference between ZnSe-QDs and ZnS-QDs.
- (3) Why the turn-on voltage (V_t) for QLEDs ($V_t \sim 1.7$ V for ZnSe-QDs based device and ~ 1.8 V for ZnS-QDs based device) is much lower than the bandgap of the QDs ($E_g \sim 2$ eV because the EL emission peak is at 631 nm).

- (4) Since the surface potential difference measured by SKPM, shown in Extended Data Figure 5, exhibits some variations, it is recommended to specify an error range for the vacuum level shift between TFB/QDs interface in Figure 1 b.
- (5) In Extended Data Figure 7 b, the axis labels should be exchanged: The horizontal axis label should be “Log (Lo)” and the vertical axis label should be “Log (T95)”
- (6) In Extended Data Figure 4 d, the energy values for TFB should be negative to make them consistent with the values for ZnS- or ZnSe-QDs.

Reviewer #2 (Remarks to the Author):

The basic theme of the paper is very interesting and has potential towards commercialization of quantum dot based LEDs. The paper should be published with some corrections.

Authors have explained their results on device performance by recording ultra-violet photoelectron spectroscopic measurements and scanning Kelvin probe microscopy. Energy levels and the wave functions are determined, and shown in Extended data Figure 5. The determination of energy levels in this fashion is not very obvious. Discussion should be extended further. Limitations of the methods used should be mentioned in the manuscript.

Role of organic capping agents on surface of quantum dots in energy level alignment is disregarded. Perhaps, the thickness of pure ZnS and ZnSe layer and organic layer may be same. Taking account of these factors is required. Authors may want to incorporate proper references of the work carried out on energy levels of graded core/shell quantum dots.

How authors determined the wave functions?

Reply to Referees' Comments

Reviewer #1 (Remarks to the Author):

In this work, Cao et al. presents a new strategy for improving hole injection by synthesizing CdSe/Cd_{1-x}Zn_xSe/ZnSe QDs (ZnSe-QDs) with low bandgap ZnSe shell instead of conventional shell of wider bandgap ZnS (ZnS-QDs). They demonstrated that the energy barrier between organic hole-transport layer (HTL) of TFB and ZnSe-QDs is lower compared to the case of the ZnS-QDs, resulting in improved hole injection in quantum dot light emitting diodes (QLEDs). As a result of improved hole injection, the QLEDs with ZnSe-QDs exhibited a maximum EQE > ~ 15% and a T95 operation lifetime > 2,300 hours at an initial brightness of 1,000 cd/m². This is a remarkable achievement toward developing stable QLEDs for practical displays. Therefore, I recommend its publication after addressing the following issues.

(1) Although the authors report record device lifetime, there is not a deep investigation on physical processes that can lead to this improvement and the degradation mechanisms for QLEDs. The authors claim that the hole injection and transport is the major factor affecting the operation lifetime for QLEDs mainly based on faster voltage rise for the ZnS-QDs based hole-only device compared to the ZnSe-QDs based hole-only device (Fig. 3 a). Even though the HTL/QD interfacial properties play a role for device degradation, there are other possible important factors, for example, unbalanced charge injection can increase non-radiative Auger-type recombination, or ZnS-QDs may have more surface defects compared to ZnSe-QDs, etc. Moreover, the QLEDs and hole-only device have different interface at the cathode. Thus, the authors are recommended to elaborate more how the ZnSe-QD-based device shows such a large enhancement in the operation lifetime.

It is difficult to directly detect the surface defects of quantum dots, however photoluminescence quantum yield (PLQY) measurement is one of the methods that can partially reveal the effects of surface defects. In our work, the ZnSe-QDs show similar PLQY (84%) as that of the ZnS-QDs (87%) in solutions, which means that the ZnS-QDs and ZnSe-QDs have comparable surface defects.

We agree that the hole only device have different interface at the cathode. It is not fair to directly compare the current of hole-only and electron-only devices, since different blocking layers are needed to blocking the opposite charges in single-carrier devices, which is not necessary in dual-carrier devices. However, it is provable to compare among all the hole-only devices or all the electron-only devices based on different QDs layers.

In QLEDs with ZnO as electron transport layers, ZnO can provide effective electrons and the hole injections are not sufficient enough to make the balance. As shown in the hole-only devices in this work, the ZnSe-QDs based hole-only device has higher current than the ZnS-QDs based device. Therefore, the ZnSe-QDs based devices have more balanced charge injections compared to the ZnS-QDs based devices, leading to less non-radiative Auger recombination, which is mainly reflected in the external quantum efficiency of the devices (15.1% vs 11.4%).

(2) According to the energy band diagrams with Fermi-level alignment at TFB/ZnS-QDs and

TFB/ZnSe-QDs interfaces in Fig. 1, the CBM of the ZnSe-QDs appears to be higher than that of the ZnS-QDs. Moreover, the shell thickness of ZnSe-QDs is larger because the core size of two QDs is the same and the diameter of the ZnSe-QDs is larger ($d \sim 9.8$ nm compared to 8.4 nm for ZnS-QDs). Thus, it is expected that less current flows in the electron-only device based on ZnSe-QDs. However, the Extended Data Figure 8 shows higher current flow for the electron-only device with ZnSe-QDs. Thus, morphology of QDs layer may be different for each case. The presence of pinholes or voids in such a thin (~ 20 nm) QDs film may affect the current density for electron-only as well as hole-only devices. Thus, the authors should address the thin film morphology difference between ZnSe-QDs and ZnS-QDs.

We conducted AFM measurement to compare the surface morphology of ZnSe-QDs and ZnS-QDs. As compared in Supplementary Figure 9, there is no significant difference in surface roughness for the two films. Therefore, the presence of pinholes or voids is not the main reason that affects the device current.

According to the references (e.g. *Opt. Mater. Express* **2**, 594-628 (2012)), the CBM of the ZnSe-QDs should be lower than that of the ZnS shell. Therefore, it is expected that the ZnSe-QDs based electron-only device show relatively higher current density than the ZnS-QDs based electron-only device.

(3) Why the turn-on voltage (V_t) for QLEDs ($V_t \sim 1.7$ V for ZnSe-QDs based device and ~ 1.8 V for ZnS-QDs based device) is much lower than the bandgap of the QDs ($E_g \sim 2$ eV because the EL emission peak is at 631 nm).

For most of the light-emitting diodes, the turn-on voltage is generally equal to or greater than the bandgap voltage of emissive layers. However, for the devices with a heterojunction of QDs and ZnO nanoparticles, sub-bandgap electroluminescence can be observed due to an Auger-assisted energy up-conversion process at the QDs/ZnO interface. Such sub-bandgap turn-on voltage has been observed and explained in our previous publications. (*Nano Today* **5**, 384-389 (2010); *Nature Photon.* **5**, 543-548 (2011))

(4) Since the surface potential difference measured by SKPM, shown in Extended Data Figure 5, exhibits some variations, it is recommended to specify an error range for the vacuum level shift between TFB/QDs interface in Figure 1 b.

We have already added error ranges for the vacuum level shift.

(5) In Extended Data Figure 7 b, the axis labels should be exchanged: The horizontal axis label should be “Log (Lo)” and the vertical axis label should be “Log (T95)”

We have already changed the axis labels of Supplementary Figure 7b.

(6) In Extended Data Figure 4 d, the energy values for TFB should be negative to make them consistent with the values for ZnS- or ZnSe-QDs.

We have already changed the energy values of TFB to be negative.

Reviewer #2 (Remarks to the Author):

The basic theme of the paper is very interesting and has potential towards commercialization of quantum dot based LEDs. The paper should be published with some corrections.

Authors have explained their results on device performance by recording ultra-violet photoelectron spectroscopic measurements and scanning Kelvin probe microscopy. Energy levels and the wave functions are determined, and shown in Extended data Figure 5. The determination of energy levels in this fashion is not very obvious. Discussion should be extended further. Limitations of the methods used should be mentioned in the manuscript.

Role of organic capping agents on surface of quantum dots in energy level alignment is disregarded. Perhaps, the thickness of pure ZnS and ZnSe layer and organic layer may be same. Taking account of these factors is required. Authors may want to incorporate proper references of the work carried out on energy levels of graded core/shell quantum dots.

We totally agree that the organic capping ligands on surface of quantum dots will affect the energy level alignment in devices. However, it is comprehensive to directly measure the energy levels of graded core/shell quantum dots and then to determine the energy level alignment at the interface. In our work, we combined UPS and SKPM to determine the hole injection barrier between the TFB and QD layers for the first time, which includes the energy level information of the ligands and a few nanometers of the shells of quantum dots. (The detect depth of UPS is about 5 nanometers and the capping ligands are about 1.5 nm in length.) We added more detailed discussions about how to determine the energy levels by using this method.

How authors determined the wave functions?

We didn't actually determine the wave functions in this work. Supplementary Figure 1a just schematically illustrates the spatial wavefunction distributions for the two quantum dots, which are drawn according to the references. In order to eliminate any misleading for the readers, we just removed these lines from the figures in the re-submitted version.

REVIEWERS' COMMENTS:

Reviewer #1 (Remarks to the Author):

Cao et al. reported the development of stable QLEDs (T95 operation lifetime > 2,300 hours at an initial brightness of 1,000 nit) by tailoring QD structure and improving hole injection into QD emissive layer. This is an important achievement towards the commercialization of the QLEDs, which is considered as a next-generation display technology after OLEDs.

They revised the manuscript appropriately by addressing most of issues raised by the referees. Therefore, I recommend its publication.

Reviewer #2 (Remarks to the Author):

The authors have answered questions satisfactorily. Authors should however, add few lines and references regarding published literature on determination of energy levels and wave functions in graded quantum dots.

Paper may be published.

Reply to Referees' Comments

Reviewer #1 (Remarks to the Author):

Cao et al. reported the development of stable QLEDs (T95 operation lifetime > 2,300 hours at an initial brightness of 1,000 nit) by tailoring QD structure and improving hole injection into QD emissive layer. This is an important achievement towards the commercialization of the QLEDs, which is considered as a next-generation display technology after OLEDs.

They revised the manuscript appropriately by addressing most of issues raised by the referees. Therefore, I recommend its publication.

Reviewer #2 (Remarks to the Author):

The authors have answered questions satisfactorily. Authors should however, add few lines and references regarding published literature on determination of energy levels and wave functions in graded quantum dots.

Paper may be published.

The description of the determination of energy levels is included in the supplementary information. We also added a few references regarding such determination both in the main text and the supplementary information.